# Research on the Effect of Urbanization on China's Carbon Emission Efficiency

**Lianshui Li [1], Yang Cai [1] and Liang Liu [2,\*]**

1  School of Applied Meteorology, Nanjing University of Information Science & Technology, Nanjing 210044, China; llsh@nuist.edu.cn (L.L.); caiyang@nuist.edu.cn (Y.C.)
2  School of Economics and Management, Southeast University, Nanjing 210096, China
\*  Correspondence: liuliang0204@seu.edu.cn

**Abstract:** Improvements in carbon emission efficiency are crucial to China's economic growth; carbon emission reduction and urbanization are two of the focuses of research on carbon emission efficiency. This paper selects 2000–2015 panel data from 30 provinces in China, evaluates the carbon emission efficiency of each province using the DEA method and, based on the STIRPAT expansion form, empirically looks at the effect of urbanization on carbon emission efficiency. The results show that, during the chosen time frame, not only did the carbon emission efficiency of China's provinces show an upward trend but the carbon emission efficiency of the Eastern, Central and Western regions differed markedly, with the highest efficiency in the Eastern region, the second highest in the Central region and the lowest in the Western region. After controlling for population density, economic development level, energy intensity and industrial structure, urbanization we determine that urbanization can indeed improve carbon emission efficiency, although there are regional differences. Urbanization is conducive to improvements in carbon emission efficiency in both the Central and Western regions but the promotion effect of the Western region is stronger. The effect in the Eastern region is not significant. Based on the conclusions above, this paper puts forward policy recommendations that promote both China's lower carbon efficiency and future environmental protection.

**Keywords:** urbanization; carbon emission efficiency; DEA; dynamic spatial panel

## 1. Introduction

With the rapid economic growth of China in recent years, environmental problems are now becoming serious issues. This is especially true of carbon emissions. Carbon emissions are shorthand for greenhouse gas emissions, which may constitute a disaster, not only for humans, but also for the planet. Such catastrophic events include, inter alia, climate anomalies, shrinking glaciers, rising sea levels caused by melting glaciers, flooding of some coastal cities, melting permafrost, loss of biodiversity. Carbon emissions are closely related to various human activities, including both daily activities and factory production. China's GDP in 2017 ranked second in the world, accounting for about 15% of world GDP. According to the 2017 Global Carbon Budget Report, made public during the Bonn Climate Conference [1], 28% of the global carbon emissions are attributable to China. In 2015, China submitted to the Secretariat of the United Nations Framework Convention on Climate Change (INDC), a report reiterating that China's carbon emissions would peak around 2030 and presenting up to 20% of autonomous goals. Economic growth and population size are the two most important factors influencing carbon emissions [2]. However, as both the world's largest developing country and the most populous country, China needs to focus on economic development. The increasing aging of the population in China may further increase carbon emissions, thereby posing an obstacle to the goal of emissions reduction [3,4]. At present, substitution of fossil fuels by renewable energy is

one of the important methods undertaken by China to achieve a reduction in carbon emissions [5]. However, China's leading role in renewable energy notwithstanding, the overall investment efficiency of the new energy industry is relatively low [6]. In addition, some evidence has come to light that environment regulation is one of the influencing factors. Nevertheless, as the current environmental policy in China stands, the goal of peak carbon emissions may not be achieved on time [7]. How to save energy and reduce carbon emissions has become an increasing concern to both the Chinese people and the government. It is urgent to solve the problem of China's carbon emissions while sustaining the demands of economic growth and improving carbon emission efficiency in a way that does not affect normal economic activities [8]. The improvement of CO2 emission efficiency is of paramount importance in reducing carbon emissions and achieving low-carbon development [9]. As China's carbon emission efficiency is still relatively low compared with developed countries, improving carbon emission efficiency is completely in line with national goals. All in all, the key to China's low-carbon economy is to improve carbon emissions efficiency; this will not only promote the sustainable development of China's economy and society but will also help optimize the world environment from an ecological and climate point of view.

Today, cities are one of the most important factors influencing carbon emissions with some studies even showing that urbanization and urbanized populations are the major drivers of carbon emissions [10,11]. Since the implementation of reforms and opening up, China's urbanization rate has gradually risen from 17.92% in 1978 to 57.35% in 2016, which is a significant increase, but still far from the 70% urbanization level and carbon emission efficiency found in developed countries. Therefore, studying the impact of urbanization on carbon emission efficiency is of great importance in pointing the way to future urbanization in China. This article studies how urbanization affects the efficiency of China's carbon emissions and computes the efficiency of carbon emissions vis a vis both China's current urbanization level and available data. This points the way not only to, how to arrange the deployment of energy conservation but also, how to formulate measures to reduce the carbon emissions and improve the efficiency of carbon. These measures will help achieve future sustainable economic and social development in China.

## 2. Literature Review

There is little in the literature on urbanization and carbon emission efficiency; relevant research mainly focuses on urbanization and carbon emissions. Even though many scholars have studied the impact of urbanization on carbon emissions and their conclusions vary a lot, there are, in essence, three main views.

The first view is that urbanization increases carbon emissions. As urbanization improves, the demand for urban infrastructure, transportation and personal consumption increases. This will lead to urban traffic congestion and overcrowding, creating, in turn, more air pollution and carbon emissions. Ren et al. pointed out that urbanization had a significant positive impact on carbon emissions; in their 25-year study of data from Shandong Province (China), they found the effect was more significant in the middle stages than in the early stages [12]. Based on the data from 216 municipal areas in China, Lu used the STIRPAT model and determined that urbanization leads to an increase in energy consumption and carbon emissions [13]. Behera and Dash based their study on 32 years of data from 17 countries in South and East Asia and used the Pedroni co-integration test to show a co-integration relationship between urbanization, energy consumption, FDI and carbon dioxide emissions [14]. Franco, et al. used 110 years of census data from India and found that the annual increase in urban population led to a faster growth in carbon dioxide emissions [15]. Fang and Tao selected nine energy types, including coal, coke and gasoline, took 15 years of urban population data in China and then used the LMDI decomposition method to prove that urbanization rate and population size do have a positive impact on carbon emissions [16].

The second view is that urbanization reduces carbon emissions. When the level of urbanization improves, urban land use will increase, public infrastructure and transportation will be improved

and car use will be reduced. Meanwhile, environmental regulations and technological innovation will reduce carbon emissions. Zhang and Xu took 10 years of data from China as a sample and adopted the STIRPAT model to empirically test if the urbanization of the land and the economy had significantly impacted carbon emissions; they found the urbanization rate of land tended to reduce carbon emissions [17]. Ali, et al. took 45 years of Singapore data and used an auto-regressive distributed lag model to conclude that urbanization inhibited the impact of carbon emissions in Singapore and that urbanization improved environmental quality by reducing carbon emissions [18]. Zhang et al. based their study on 13 years of panel data in China and, using three estimation methods, including the fixed-effect model, showed that urbanization reduces China's carbon emissions through human capital accumulation and clean production [19]. Based on data over 25-years from ten Asian countries, including China and India, Bilgili et al. [20] used both unit root and co-integration tests to prove that urbanization has a negative impact on carbon emissions.

The third view is that urbanization and carbon emissions are not simply linear. When the level of urbanization is low, a huge change in the pattern of consumption occurs with improvements in urbanization; this puts strain on various public facilities. Energy consumption will increase rapidly and carbon emissions will increase. When the level of urbanization is high, the level of economic development is even higher than that of the already high consumption. At the same time, a low-carbon consciousness will emerge so it could be argued that urbanization plays a moderating role in carbon emissions. Martini-Zarzoso and Maruotti used data from 88 developing countries to demonstrate an inverted U-shaped relationship between urbanization and carbon dioxide emissions [21]. Cao et al. based their study on 36 years data after China's reform and opening up and, adopting a threshold regression, concluded that the relationship between urbanization and $CO_2$ emissions showed periodicity and regional characteristics [22]. As an example, Wang et al. used data from 30 provinces and cities in China over the past 15 years and, employing STIRPAT, proved that urbanization had a positive effect on carbon emissions in Western China, a negative effect on carbon emissions in Central China and a non- statistically significant effect in Eastern China [23]. He et al. based their study on 19 years' provincial panel data in China and, using the STIRPAT model as an example, verified that the impact of urbanization varies greatly in different regions [24].

The literature focuses on the effect of urbanization on carbon emissions and draws rich conclusions. This does indeed provide some reference for the study of this article. However, there is still little literature on the effect of urbanization on carbon emission efficiency. In the context of China's economic growth and the need for reduction in carbon emissions, improving carbon emission efficiency is of great importance. At the same time, an analysis of the impact of urbanization on carbon emission efficiency is certainly worthwhile as it can serve to guide the development of China's new urbanization. Compared to existing research, the innovations of this paper are as follows: first, this paper studies the relationship between urbanization and carbon emission efficiency, an important complement to the literature. Second, using both the static and dynamic spatial models, we estimate the space overflow effect of urbanization on carbon emission efficiency. Third, taking the case of China's 30 provinces (cities), we compare the heterogeneous impacts of urbanization scale; this is important for both the formulation of urbanization development policies and science-based urban development in China.

## 3. Carbon Emission Efficiency Measurement and Result Analysis

### 3.1. The SBM Model

This paper uses the DEA SBM model to calculate the carbon emission efficiency of each province. The DEA method is a model established by Charnes et al. to evaluate the efficiency of multiple decision making units with multiple inputs and outputs [25]. DEA is a kind of non-parametric method, which does not need a specific function form but only needs specific output value. Tone modified the basic model of DEA, improved the problem of input-output relaxation and proposed the SBM model [26]. Because the traditional DEA model cannot examine all the relaxation variables, there will be deviations

in the efficiency evaluation. The relaxation variable problem can be solved by the SBM model. At the same time, the measurement values of the SBM model are less than those of the traditional DEA model, which makes the comparison of each decision making unit (DMU) more convenient. The specific model is as follows:

$$\min\rho = \left[1 - (1/m)\sum_{i=1}^{m} s_i^- / x_{i0}\right] / \left[1 - (1/s)\sum_{i=1}^{s} s_i^+ / y_{r0}\right]$$
$$x_0 = X\lambda + s^-$$
$$y_0 = Y\lambda - s^+$$
$$\lambda \geq 0; s^+ \geq 0; s^+ \geq 0$$

(1)

In Equation (1), $\rho$ is the value of efficiency, $x_0$ and $y_0$ are the input and output of the DMU, $x_{i0}$ and $y_{r0}$ are the elements of $x_0$ and $y_0$, x and y is the DMU input-output matrix, $\lambda$ is the weight column matrix. When $0 < \rho < 1$, the DMU is not valid, and when $\rho = 1$, the DMU is valid.

### 3.2. Research Indicators and Data Selection

As there is no authoritative carbon emission data in China, the carbon emission data in this paper are calculated. Considering the consumption of fossil fuels is the main source of carbon emissions, this paper selects eight kinds of fossil fuels–coal, coke, crude oil, kerosene, gasoline, diesel, fuel oil and natural gas. The annual consumption of these eight kinds of fossil fuels comes from the China energy statistical yearbook. We then use the carbon emission calculation method provided by the IPCC to calculate carbon emissions. The formula can be shown as follows:

$$(CO_2)_t = \sum_{i=1}^{8} (CO_2)_{it} = \sum_{i=1}^{8} E_{it} \times NCV_{it} \times CEF_{it} \times COF_{it} \times 44/12$$

(2)

In Equation (2), $CO_2$ is the carbon emission, $i$ is the type of fossil fuel, $t$ is time, $E$ is fuel consumption, NCV is low calorific value, CEF is carbon content, COF is oxidation rate of carbon, 44 and 12 are the molecular weights of carbon dioxide and carbon respectively. We can then calculate the carbon emission coefficients of various fossil fuels by NCV, CEF and COF.

Carbon emissions and GDP are counted as outputs; the input variables are capital, labor and energy. Capital refers to the net value of fixed assets, labor to the number of employees and energy to the total energy consumption. Provincial and urban labor force data from 2000 to 2008 were obtained from a compilation of 60 years of the new China statistical data and the labor force data of provinces and cities from 2009 to 2016 were obtained from the annual statistical yearbook of each province. The GDP and capital of all provinces and cities come from the China statistical yearbook and the energy data of all provinces and cities come from the China Energy Statistical Yearbook. Descriptive statistics of study samples are shown in Table 1.

**Table 1.** Descriptive analysis of sample data.

| Categories | Variables | The Proxy Variables | Unit | Max | Min | Standard Deviation | The Mean |
|---|---|---|---|---|---|---|---|
| Input | Capital | Net fixed assets | Hundred million RMB | 35,587.4 | 160.46 | 6791.101 | 6751.503 |
| | Labour | The number of jobs | Ten thousand people | 6726.0 | 275.5 | 1679.449 | 2495.408 |
| | Energy | Total energy consumption | Ten thousand tons standard coal | 85,857.509 | 399.360 | 9720.539 | 11,892.351 |
| Output | GDP | —— | One hundred million RMB | 80,854.91 | 263.59 | 13,346.559 | 12,507.752 |
| | CO2 | —— | Ten thousand tons of | 54,295.632 | 239.269 | 6868.665 | 8361.558 |

### 3.3. Results Analysis of Carbon Emission Efficiency

With the SBM model of DEA and the input-output index in Table 1 as the basis, EMS1.3 software was used to process the above data. The carbon emission efficiency values of 30 provinces in China from 2000 to 2015 is shown in Table 2.

**Table 2.** Carbon emission efficiency of 30 provinces in China from 2000 to 2015.

| Year | 2000 | 2001 | 2002 | 2003 | 2004 | 2005 | 2006 | 2007 | 2008 | 2009 | 2010 | 2011 | 2012 | 2013 | 2014 | 2015 |
|---|---|---|---|---|---|---|---|---|---|---|---|---|---|---|---|---|
| Beijing | 0.097 | 0.102 | 0.124 | 0.154 | 0.190 | 0.274 | 0.175 | 0.209 | 0.307 | 0.293 | 0.358 | 0.515 | 0.498 | 0.810 | 1.000 | 1.000 |
| Tianjin | 0.120 | 0.126 | 0.154 | 0.183 | 0.187 | 0.220 | 0.255 | 0.262 | 0.359 | 0.282 | 0.541 | 0.748 | 0.747 | 0.930 | 1.000 | 1.000 |
| Hebei | 0.040 | 0.037 | 0.039 | 0.043 | 0.045 | 0.048 | 0.053 | 0.064 | 0.075 | 0.078 | 0.111 | 0.156 | 0.154 | 0.250 | 0.198 | 0.239 |
| Liaoning | 0.059 | 0.060 | 0.070 | 0.084 | 0.097 | 0.113 | 0.137 | 0.136 | 0.179 | 0.176 | 0.226 | 0.301 | 0.356 | 0.468 | 0.365 | 0.435 |
| Shanghai | 0.165 | 0.175 | 0.226 | 0.298 | 0.311 | 0.323 | 0.391 | 0.477 | 0.556 | 0.499 | 0.802 | 0.932 | 0.956 | 0.911 | 0.951 | 0.889 |
| Jiangsu | 0.143 | 0.169 | 0.202 | 0.243 | 0.228 | 0.232 | 0.314 | 0.421 | 0.489 | 0.439 | 0.627 | 0.710 | 0.849 | 1.000 | 1.000 | 1.000 |
| Zhejiang | 0.152 | 0.173 | 0.190 | 0.247 | 0.222 | 0.265 | 0.303 | 0.360 | 0.397 | 0.386 | 0.703 | 0.780 | 0.799 | 0.631 | 0.633 | 0.583 |
| Fujian | 0.110 | 0.126 | 0.134 | 0.157 | 0.172 | 0.138 | 0.154 | 0.179 | 0.210 | 0.199 | 0.276 | 0.478 | 0.413 | 0.386 | 0.431 | 0.609 |
| Shandong | 0.128 | 0.124 | 0.145 | 0.132 | 0.154 | 0.192 | 0.266 | 0.369 | 0.452 | 0.477 | 0.571 | 0.727 | 0.811 | 1.000 | 1.000 | 1.000 |
| Guangdong | 0.205 | 0.218 | 0.244 | 0.235 | 0.341 | 0.364 | 0.369 | 0.506 | 0.602 | 0.475 | 0.735 | 0.914 | 0.739 | 1.000 | 1.000 | 1.000 |
| Hainan | 0.082 | 0.079 | 0.064 | 0.055 | 0.041 | 0.084 | 0.117 | 0.159 | 0.167 | 0.136 | 0.226 | 0.218 | 0.214 | 0.175 | 0.185 | 0.184 |
| Shanxi | 0.053 | 0.042 | 0.046 | 0.058 | 0.078 | 0.114 | 0.129 | 0.162 | 0.088 | 0.098 | 0.189 | 0.284 | 0.292 | 0.346 | 0.301 | 0.352 |
| Jilin | 0.140 | 0.159 | 0.190 | 0.196 | 0.212 | 0.192 | 0.189 | 0.228 | 0.323 | 0.360 | 0.191 | 0.459 | 0.381 | 0.485 | 0.546 | 0.543 |
| Heilongjiang | 0.235 | 0.242 | 0.235 | 0.277 | 0.326 | 0.418 | 0.399 | 0.422 | 0.266 | 0.233 | 0.295 | 0.317 | 0.648 | 0.393 | 0.309 | 0.377 |
| Anhui | 0.073 | 0.070 | 0.082 | 0.099 | 0.152 | 0.188 | 0.214 | 0.264 | 0.163 | 0.225 | 0.382 | 0.541 | 0.646 | 0.622 | 0.632 | 0.424 |
| Jiangxi | 0.115 | 0.117 | 0.151 | 0.175 | 0.196 | 0.251 | 0.277 | 0.369 | 0.204 | 0.271 | 0.468 | 0.639 | 0.753 | 1.000 | 1.000 | 0.931 |
| Henan | 0.141 | 0.145 | 0.162 | 0.185 | 0.184 | 0.189 | 0.215 | 0.268 | 0.357 | 0.380 | 0.377 | 0.574 | 0.734 | 0.645 | 0.599 | 0.630 |
| Hubei | 0.107 | 0.112 | 0.122 | 0.131 | 0.153 | 0.165 | 0.189 | 0.220 | 0.133 | 0.171 | 0.289 | 0.343 | 0.432 | 0.695 | 0.606 | 0.648 |
| Hunan | 0.127 | 0.101 | 0.110 | 0.138 | 0.158 | 0.124 | 0.155 | 0.199 | 0.133 | 0.189 | 0.366 | 0.591 | 0.613 | 0.747 | 0.631 | 0.627 |
| Neimenggu | 0.037 | 0.034 | 0.038 | 0.037 | 0.043 | 0.053 | 0.070 | 0.089 | 0.104 | 0.114 | 0.159 | 0.245 | 0.167 | 0.214 | 0.176 | 0.195 |
| Guangxi | 0.052 | 0.048 | 0.060 | 0.053 | 0.063 | 0.071 | 0.076 | 0.088 | 0.109 | 0.116 | 0.153 | 0.191 | 0.234 | 0.259 | 0.298 | 0.351 |
| Chongqing | 0.043 | 0.052 | 0.054 | 0.082 | 0.088 | 0.079 | 0.102 | 0.137 | 0.113 | 0.121 | 0.196 | 0.226 | 0.247 | 0.409 | 0.441 | 0.486 |
| Sichuan | 0.068 | 0.070 | 0.073 | 0.061 | 0.072 | 0.134 | 0.116 | 0.136 | 0.137 | 0.161 | 0.200 | 0.255 | 0.267 | 0.283 | 0.359 | 0.331 |
| Quizhou | 0.040 | 0.039 | 0.043 | 0.036 | 0.034 | 0.030 | 0.042 | 0.054 | 0.071 | 0.076 | 0.087 | 0.096 | 0.098 | 0.160 | 0.219 | 0.222 |
| Yunnan | 0.069 | 0.071 | 0.065 | 0.061 | 0.092 | 0.048 | 0.063 | 0.074 | 0.076 | 0.075 | 0.091 | 0.105 | 0.113 | 0.105 | 0.113 | 0.125 |
| Shanxi | 0.083 | 0.071 | 0.077 | 0.090 | 0.094 | 0.096 | 0.117 | 0.127 | 0.139 | 0.141 | 0.168 | 0.190 | 0.211 | 0.211 | 0.219 | 0.225 |
| Gansu | 0.065 | 0.063 | 0.073 | 0.072 | 0.072 | 0.075 | 0.093 | 0.106 | 0.118 | 0.129 | 0.160 | 0.184 | 0.181 | 0.198 | 0.156 | 0.164 |
| Qinghai | 0.062 | 0.060 | 0.074 | 0.066 | 0.074 | 0.093 | 0.086 | 0.094 | 0.091 | 0.092 | 0.150 | 0.150 | 0.112 | 0.078 | 0.074 | 0.086 |
| Ningxia | 0.032 | 0.023 | 0.019 | 0.016 | 0.041 | 0.038 | 0.059 | 0.078 | 0.070 | 0.079 | 0.094 | 0.089 | 0.109 | 0.086 | 0.067 | 0.071 |
| Xinjiang | 0.065 | 0.071 | 0.076 | 0.079 | 0.083 | 0.086 | 0.082 | 0.092 | 0.096 | 0.081 | 0.101 | 0.109 | 0.087 | 0.102 | 0.102 | 0.112 |

The 30 provinces in China are divided into three regions: the Eastern, the Central and the Western (Tibet, Hong Kong, Macao and Taiwan lack energy data with which to calculate the total carbon emission, so they are omitted from the study). It can be seen from Figures 1–3 that the carbon emission efficiency of the three regions in China show a rising trend line. The highest carbon emission efficiency of the Eastern and Central regions is 1, while the highest carbon emission efficiency of the Western regions is 0.6. It can be clearly seen that the carbon emission efficiency in the Western regions is far lower than that of the Eastern and Central regions. However, from 2000 to 2015, Hebei's carbon emission efficiency and its growth rate were far lower than that of other provinces of the Eastern region, just slightly more than that for Hainan.

The reason may be that the coal and metallurgical industries, which are important industries in Hebei, exhibit low carbon emission efficiency. In addition, according to "The Outline of Coordinated Development for the Beijing-Tianjin-Hebei region", some basic industries, notably the high pollution industries, will continue to relocate to Hebei, meaning that Hebei will face more pressure on carbon emissions and therefore have even more environmental problems [27].

Figure 4 shows the annual carbon emission efficiency of each province, calculated by DEA, and then calculated and analyzed by regions. On the whole, both the nation as a whole and the three sub-regions show a similar trend; the overall trend of carbon emission efficiency is rising. The Eastern region has the highest carbon emission efficiency, followed by the Central and Western regions.

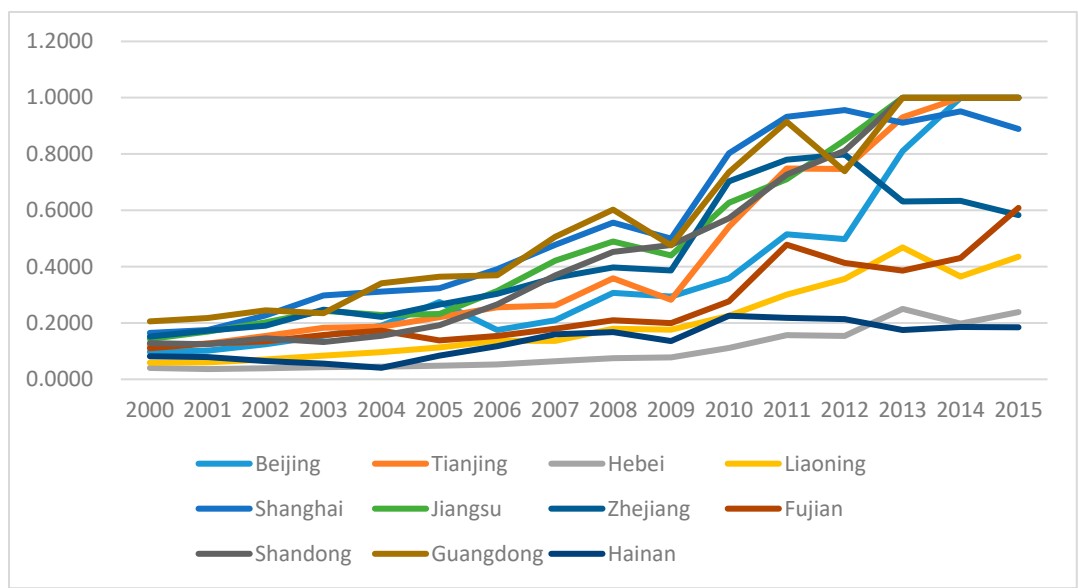

**Figure 1.** Carbon emission efficiency of provinces and cities in eastern China from 2000 to 2015.

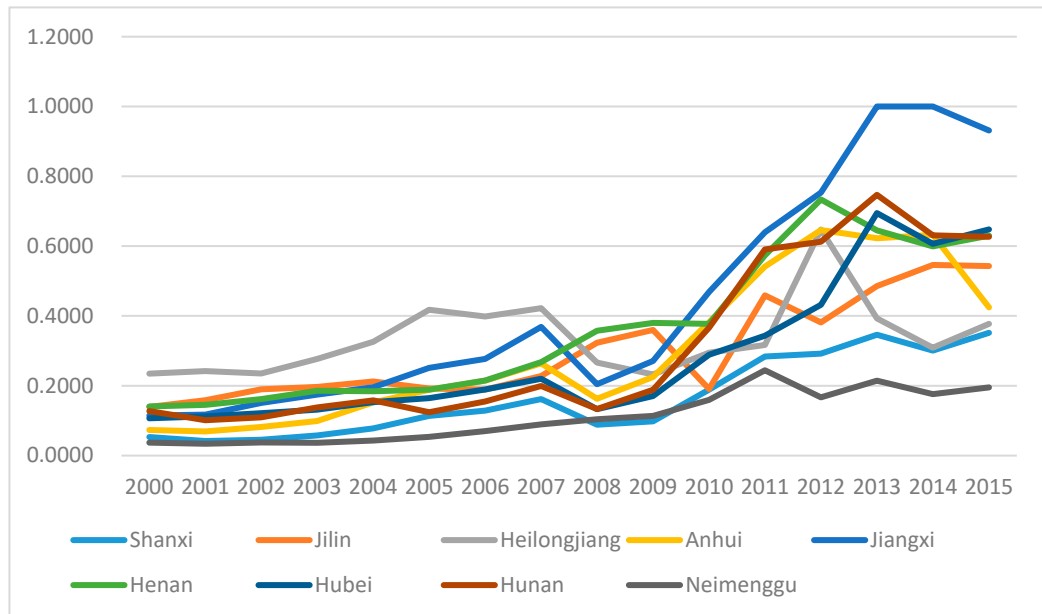

**Figure 2.** Carbon emission efficiency of provinces and cities in central China from 2000 to 2015.

In particular, the nation and the three sub-regions showed a gradual increase in carbon efficiency between 2000 and 2007. During this period, China put forward its tenth and eleventh "five-year plans", both of which incorporate ecological improvements, environmental protection and environmental governance into one of the key special plans. To this end, the government has enacted relevant measures and laws and the carbon emission efficiency has gradually increased.

From 2007 to 2009, the efficiency of carbon emissions in the Eastern and Central regions declined by varying degrees while that of the Western region was basically flat. In 2007, as the financial crisis quickly swept across the world, China exhibited a negative increase in GDP. In response to the economic crisis, China proposed a series of measures, including the speeding up of the construction of infrastructure such as railways and roads. As a result, some secondary industry pollution-intensive enterprises again started to appear with consequent reductions in carbon emission efficiency.

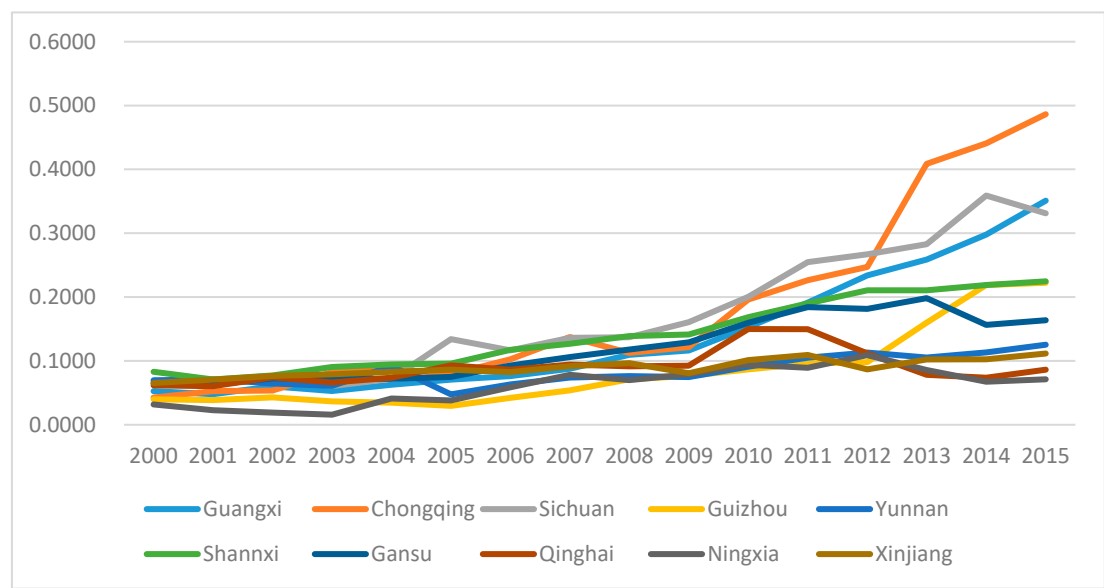

**Figure 3.** Carbon emission efficiency of provinces and cities in western China from 2000 to 2015.

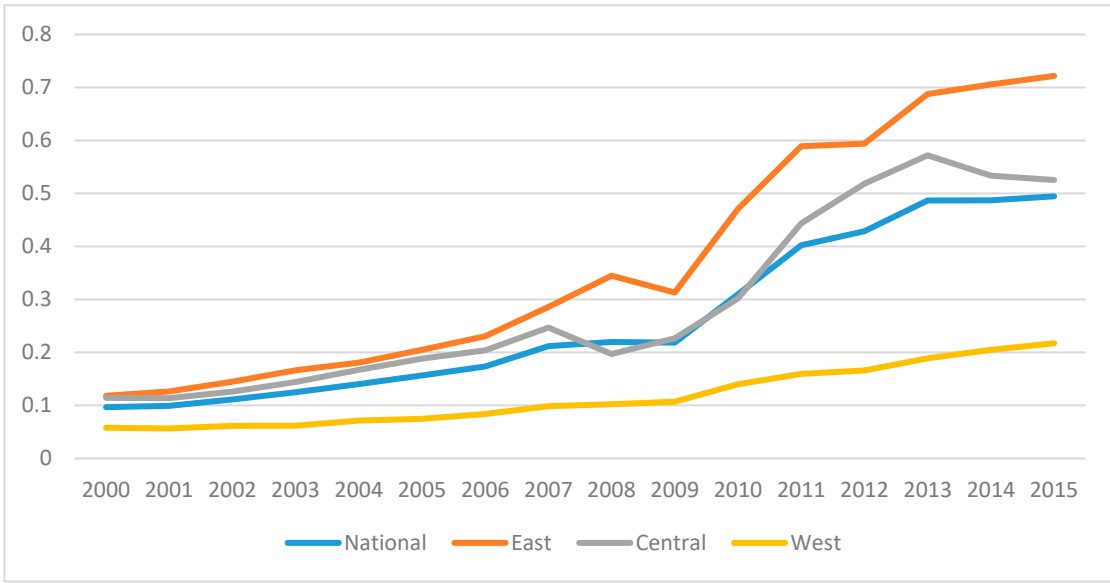

**Figure 4.** Change of carbon emission efficiency by region in China from 2000 to 2015.

After 2009, the carbon emission efficiency of China's Eastern, Central and Western regions increased rapidly. Because of China's commitments at the World Meteorological conference in Copenhagen, the carbon emissions per unit of GDP must be reduced by 40–45% by 2020. To this end, China has taken on tackling climate change as a major strategy for national economic and social development and has sought to constantly improve carbon emission efficiency through industrial restructuring and technological progress.

In Table 3 the carbon emission efficiency of each province from 2000 to 2015 calculated by the DEA model, with calculated average values is given. The carbon emission efficiency of each province is assigned by region, Eastern, Central or Western region, and then ranked. As can be seen from Table 3, the carbon emission efficiency of the Eastern region is higher than that of the Central region, while that of the Western region is the lowest.

**Table 3.** Average carbon emission efficiency of each province from 2000 to 2015.

| The Eastern Region | Carbon Emission Efficiency | Ranking | The Central Region | Carbon Emission Efficiency | Ranking | The Western Region | Carbon Emission Efficiency | Ranking |
|---|---|---|---|---|---|---|---|---|
| Beijing | 0.382 | 8 | Shanxi | 0.164 | 19 | Inner Mongolia | 0.111 | 24 |
| Tianjin | 0.445 | 5 | Jilin | 0.299 | 12 | Guangxi | 0.139 | 21 |
| Hebei | 0.102 | 25 | Heilongjiang | 0.337 | 10 | Chongqing | 0.180 | 17 |
| Liaoning | 0.204 | 16 | Anhui | 0.299 | 13 | Sichuan | 0.170 | 18 |
| Shanghai | 0.554 | 2 | Jiangxi | 0.432 | 6 | Guizhou | 0.084 | 28 |
| Jiangsu | 0.504 | 3 | Henan | 0.362 | 9 | Yunnan | 0.084 | 29 |
| Zhejiang | 0.427 | 7 | Hubei | 0.282 | 14 | Shaanxi | 0.141 | 20 |
| Fujian | 0.261 | 15 | Hunan | 0.313 | 11 | Gansu | 0.119 | 23 |
| Shandong | 0.472 | 4 | | | | Qinghai | 0.090 | 26 |
| Guangdong | 0.559 | 1 | | | | Ningxia | 0.061 | 30 |
| Hainan | 0.137 | 22 | | | | Xinjiang | 0.089 | 27 |
| Mean | 0.368 | | | 0.311 | | | 0.115 | |

Figure 5 takes the average carbon emission data of each province and city in Table 3 so as to display the carbon emission efficiency distribution of each province more intuitively. We have ranked the average carbon efficiency from large to small, and divided the data into five intervals; the number of provinces and cities in each interval are equal. Respectively they are greater than or equal to 0.4323 for the high average carbon efficiency area, 0.2996–0.4323 for the higher average carbon efficiency area, average 0.1702–0.2995 for the medium carbon efficiency area, average 0.111–0.172 for the low carbon efficiency area; the average is less than 0.111 for the low carbon efficiency area. The darker the color, the higher the average carbon efficiency.

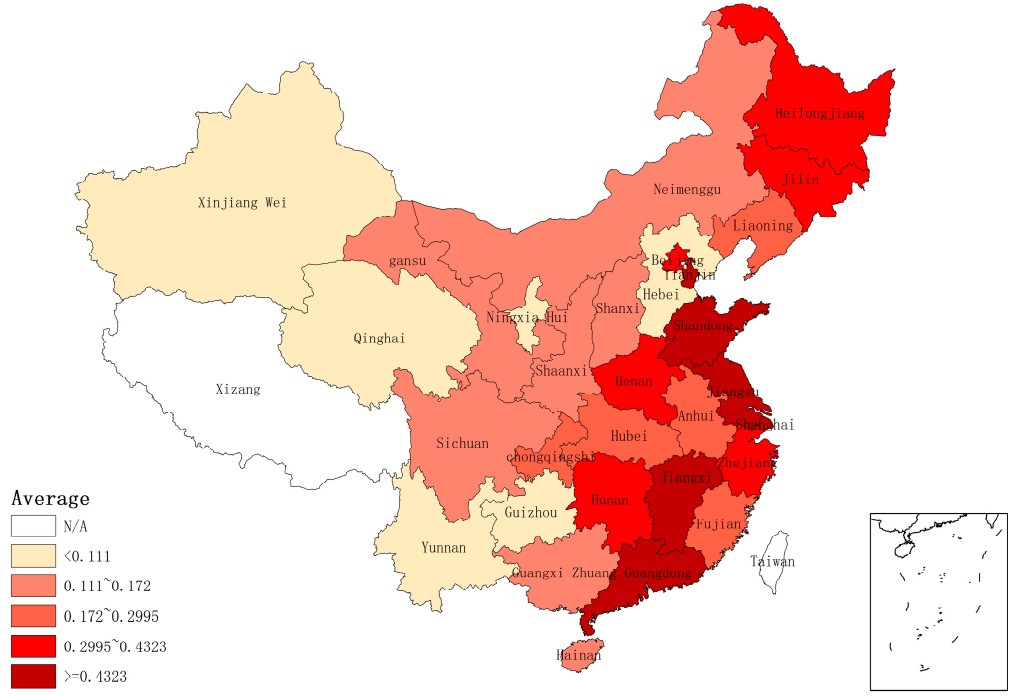

**Figure 5.** Regional distribution of average carbon emission efficiency of provinces and cities from 2000 to 2015.

Of the 11 provinces and cities in the Eastern region, all except Hebei and Hainan are in the medium or above carbon emission efficiency range. The Eastern region accounts for the majority of the high average carbon emission efficiency figures. Of the eight provinces and cities in the Central region, most are in the range of moderate carbon efficiency but of the 11 provinces and cities in the Western

region (Tibet lacks the energy data used to calculate total carbon emissions and so is omitted), only Chongqing and Sichuan are in the range of average medium carbon efficiency.

## 4. Establishment of Measurement Model and Data Description

### 4.1. Original Model

Most of the studies on the relationship between environmental problems and economic population have adopted the IPAT model proposed by Ehrlich and Holdelanno, in which I represents ecological environment problems, P represents population size, A represents economic development level and T represents technical level [28]. However, IPAT was an early model with certain limitations. For example, in IPAT the elasticity of population size, economic development level and technical level is assumed to be 1; this is not, however, the case in practice. Modeled on the IPAT, Dietz and Rosa proposed a random environmental impact assessment model, namely the STIRPAT model, to evaluate the relationship between the dependent variable and the three independent variables of population size, economic development level and technical level [29]:

$$I = aP^b A^c T^d \varepsilon \qquad (3)$$

In Equation (3), *I*, *P*, *A* and *T* are still the original indicators. *a*, *b*, *c* and *d* are the prediction parameters of the model, and $\varepsilon$ is the random error term of the model. Differing from the original IPAT model, in the STIRPAT model *T* can choose different economic variables as explanatory variables. For example, Wang and Liu adopted energy consumption intensity, i.e., the ratio of energy consumption to GDP [30]; Ma adopted R&D intensity, i.e., the ratio of R&D expenditure to GDP [31]. In this paper, *T* is the energy consumption intensity, that is, the ratio of total energy consumption to GDP. In the calculation process, the logarithm of both ends of Equation (3) is usually taken, and the specific formula is as follows:

$$\ln I = \ln a + b \ln P + c \ln T + \varepsilon \qquad (4)$$

### 4.2. Model Extension

This paper also uses the extended form of STIRPAT to study the effect of urbanization on carbon emission efficiency. Urbanization can affect carbon emission efficiency from two aspects. On the one hand, an increase in urbanization levels will lead to an enhancement in scale efficiency of public facilities and a reduction in car usage, mobile distance and power supply and deployment. At the same time, when urbanization levels are high, energy-saving products and high-tech products will increase; consumers will tend to turn to energy-saving products, thereby reducing carbon emissions. On the other hand, an increase in urbanization levels also leads to urban traffic jams, overcrowding and more air pollution. Moreover, at the same time, the increase in urbanization levels necessitates an increase in urban land area which then requires the construction of buildings and public facilities, all of which increase carbon emissions. At the same time, industrial structure index is added, also an important factor affecting carbon emission efficiency. The industrial structure here is the proportion of secondary industry GDP to inter-provincial GDP of each province. Because secondary industry is often composed of processing manufacturing industries, the processing and manufacturing operations will generate large amounts of $CO_2$, which, to a certain extent, determines the size of carbon emissions.

Urbanization (U) and industrial structure (S) were added to the model to establish the following general static panel model:

$$\ln I_{i,t} = \beta_0 + \beta_1 \ln P_{i,t} + \beta_2 \ln A_{i,t} + \beta_3 \ln T_{i,t} + \beta_4 \ln U_{i,t} + \beta_5 \ln S_{i,t} + \varepsilon_{i,t} \qquad (5)$$

Considering that in real life, changes in some of the variables will have a certain lag effect and, considering that carbon emissions are indicators that exhibit dynamic effect, the carbon dioxide emissions in this year are derived from increases or decreases of carbon emissions in the previous year.

After changes in each item, a change of each item's explained variable in the previous period usually cannot take effect immediately. They usually affect the carbon emission efficiency in the following period. Therefore, this paper has included the first-order lag item of carbon emission efficiency and established the following general dynamic panel model:

$$\ln I_{i,t} = \beta_0 + \tau \ln I_{i,t-1} + \beta_1 \ln P_{i,t} + \beta_2 \ln A_{i,t} + \beta_3 \ln T_{i,t} + \beta_4 \ln U_{i,t} + \beta_5 \ln S_{i,t} + \varepsilon_{i,t} \tag{6}$$

Considering the spatial linkage and spatial spillover of carbon emissions between regions, the carbon emission efficiency of surrounding areas may affect the carbon emission efficiency of the region through a spatial spillover effect. Therefore, this paper has included the spatial lag item of carbon emission efficiency and established the following static spatial panel model:

$$\ln I_{i,t} = \beta_0 + \rho \sum W_{i,j} \ln I_{j,t} + \beta_1 \ln P_{i,t} + \beta_2 \ln A_{i,t} + \beta_3 \ln T_{i,t} + \beta_4 \ln U_{i,t} + \beta_5 \ln S_{i,t} + \alpha_i + v_{t+} \varepsilon_{i,t}$$
$$\varepsilon_{i,t} = \lambda \sum W_{i,j} \varepsilon_{j,t} + \mu_{i,t} \tag{7}$$

In Equation (7), $\alpha_i, v_t, \varepsilon_{i,t}$ represent the regional effect, the time effect and the random disturbance term respectively, reflecting the random disturbances of different dimensions that affect carbon emission efficiency. *W* represents the spatial weight matrix, reflecting the spatial correlation between regions. In this paper, geographical distance is used to construct the spatial weight matrix, which can fully consider the fact that two non-adjacent regions in space, also interact with each other. It is possible for carbon emission efficiency to have both spatial and dynamic effects, this paper, based on Equation (5), includes both spatial lag terms and early lag terms of carbon emission efficiency and constructs the following dynamic spatial panel model:

$$\ln I_{i,t} = \beta_0 + \rho \sum W_{i,j} \ln I_{j,t} + \tau \ln I_{i,t-1} + \beta_1 \ln P_{i,t} + \beta_2 \ln A_{i,t} + \beta_3 \ln T_{i,t} + \beta_4 \ln U_{i,t} + \beta_5 \ln S_{i,t} + \alpha_i + v_t + \varepsilon_{i,t}$$
$$\varepsilon_{i,t} = \lambda \sum W_{i,j} \varepsilon_{j,t} + \mu_{i,t} \tag{8}$$

*4.3. Variable Description*

### 4.3.1. Explained Variable

Carbon emission efficiency (I): this paper selects data on eight types of energies in 30 provinces and cities in China from 2000 to 2015 and calculates the total annual inter-provincial carbon emissions according to IPCC2007. It then takes annual inter-provincial GDP, capital and labor data and uses the DEA model to calculate inter-provincial carbon emission efficiency.

### 4.3.2. Core Explanatory Variable

Urbanization level (U): ecological modernization theory, urban environment transformation theory and compact city theory show that urbanization is an important factor affecting carbon emissions, which, in turn, affect carbon emission efficiency. We use the proportion of inter-provincial urban population to the total population to measure the level of urbanization; this is expected to have a significant positive impact on carbon emission efficiency.

### 4.3.3. Control Variables

(1) Population size (P): population size is an important factor affecting carbon emissions. Carbon emissions have spatial attributes so population density is used as a proxy variable for population size in this paper. (2) Economic development level (A): according to environmental Kuznets theory, economic development level is an important factor affecting carbon emissions. This paper uses GDP per capita to measure the level of economic development. (3) Energy intensity (E): energy intensity is an important reflection of technical level and has an important impact on carbon emissions. This paper uses the ratio of total energy consumption to GDP to measure energy intensity. (4) Industrial structure (S): industry is the main source of carbon emissions, so industrial structure is also an important factor

affecting emissions. We use the proportion of secondary industry GDP to provincial GDP to measure the industrial structure.

*4.4. Data Sources*

The regional gross domestic product (GDP), per capita income level and capital amount of the provinces and cities involved in this paper all come from the China Statistical Yearbook. The labor force data of provinces and cities from 2000 to 2008 were obtained from the Compilation of 60 Years of Statistical Data of New China and the labor force data of provinces and cities from 2009 to 2016 were obtained from the annual statistical yearbook of each province. The eight energy indicators needed to calculate total new energy consumption and total carbon emissions are all from the China Energy Statistics Yearbook. The level of urbanization, GDP of secondary industry and population size of each province in China come from the annual statistical yearbook of each province. The descriptive statistics of the sample are shown in Table 4.

**Table 4.** Statistical description of variables.

| Variable Symbols | Variable Names | Sample Size | Average Value | Maximum Value | Minimum Value | Standard Deviation |
|---|---|---|---|---|---|---|
| I | Carbon emission efficiency | 480 | 0.260 | 1.000 | 0.0157 | 0.241 |
| U | Urbanization rate | 480 | 0.489 | 0.896 | 0.233 | 0.153 |
| P | Population density | 480 | 0.431 | 0.985 | 0.0517 | 0.255 |
| A | Economic development level | 480 | 0.275 | 1.080 | 0.0266 | 0.216 |
| E | Energy intensity | 480 | 1.354 | 5.229 | 0.298 | 0.826 |
| S | Industrial structure | 480 | 0.469 | 0.615 | 0.197 | 0.0766 |

## 5. Empirical Analysis

*5.1. Regression Results Analysis at the National Level*

Model (I) is a general *static* panel model which uses the feasible generalized least square method (FGLS) to estimate. Model (II) is a general dynamic panel model where the System GMM method is used to estimate. Model (III) is a static spatial panel model; we use the maximum likelihood method (ML) for estimations. Model (V) is a dynamic spatial panel model where we use the spatial GMM method for estimations. The regression results are shown in Table 5 below.

Comparing Models (I), (II), (III) and (V), we see that the regression results of model (V) are better than those of Models (I) (II) or (III). This may be because the dynamic and spatial effects of carbon emission efficiency have been considered in the regression process of model (V) making the estimation results more accurate and reliable. In the regression results of model (V), the coefficients of the time lag and spatial lag terms are significantly positive, which clearly shows that there are significant dynamic and spatial effects in carbon emission efficiency. However, if we do not consider the dynamic and spatial effects at the same time in the regression process, it may lead to biased estimations; Model V is therefore chosen as the interpretation model.

From the regression results of Model (V), we find that the urbanization coefficient is positive at the 1% significance level, indicating that an improvement in urbanization rate is conducive to an improvement in carbon emission efficiency. When the urbanization rate is increased by 1%, the carbon emission efficiency is increased by 0.931%. The possible reasons are as follows. First, urbanization is conducive to the intensive use and recycling of energy, as well as the centralized treatment of pollutants and wastes. Urbanization is also conducive to the extensive and effective use of both energy conservation and emission reduction technologies and so has a significant role in improving carbon emission efficiency. Second, with increases in urbanization level, the scale benefit of urban public facilities will be enhanced and car usage, moving distance, power supply and deployment will be reduced. This effectively reduces the carbon emissions of transportation and electricity

generation, leading to significant improvements in carbon emission efficiency. Third, when the level of urbanization is improved, the rural population is transformed into an urban population with higher levels of education. With improvements in the quality of human resources, higher quality human resources will displace other material resources. Meanwhile, more people will advocate low-carbon consumption; carbon emission efficiency will thus be improved. In terms of control variables, the coefficient of population density is significantly positive, indicating that increases in population density are conducive to improvements in carbon emission efficiency. This may be because high population density is conducive to the intensive use of energy, effectively reducing the cost of transportation and living; this has a significant role in promoting economic growth and energy efficiency. Improvements in energy intensity have a significant inhibitory effect on improvements in carbon emission efficiency. This is mainly because, when the energy intensity increases, the energy consumption brought by economic growth will increase significantly, bringing about an increase in carbon emissions. An increase in the proportion of secondary industry is not conducive to improvements in carbon emission efficiency. This is mainly because an increase in the proportion of secondary industry will also lead to a sharp increase in energy consumption, resulting in an increase in carbon emissions. Improvements in the level of economic development beyond what is expected reduce the efficiency of carbon emissions; this may be due to the fact that China is currently on the left side of the inverted "U" inflection point of the Environmental Kuznets curve. The pursuit of rapid economic growth brings about a sharp increase in energy consumption, but the technological and structural effects caused by economic growth are significantly lower than the scale effect.

**Table 5.** Regression results at the national level.

| Variables | I (FGLS) | II (Sys-GMM) | III (ML) | V (Spatial-GMM) |
|---|---|---|---|---|
| $\tau$ (time lag term) | | 0.425 *** | | 0.137 *** |
| | | (5.165) | | (3.275) |
| $\rho$ (Spatial lag term) | | | 0.325 *** | 0.014 ** |
| | | | (5.286) | (2.172) |
| urban | 0.719 *** | 0.906 *** | 0.598 * | 0.831 *** |
| | (2.697) | (3.253) | (1.731) | (3.618) |
| population | 0.233 *** | 0.216 *** | 0.278 *** | 0.237 *** |
| | (4.180) | (3.725) | (4.580) | (4.081) |
| pgdp | −0.061 * | −0.054 | −0.047 * | −0.052 ** |
| | (−1.787) | (−1.119) | (−1.812) | (−2.125) |
| structure | −0.762 | −0.981** | −0.637 | −0.707 *** |
| | (−1.306) | (−2.064) | (−1.152) | (−2.545) |
| energy | −0.482 *** | −1.106 *** | −0.966 *** | −1.003 *** |
| | (−5.279) | (−4.871) | (−5.198) | (−5.105) |
| _cons | −1.089 *** | −1.178 *** | −1.436 *** | −1.926 *** |
| | (−6.724) | (−7.104) | (−5.813) | (−7.724) |
| AR(1) | | −3.42 | | −3.86 |
| Test(p) | | (0.001) | | (0.000) |
| AR(2) | | −1.23 | | −1.28 |
| Test(p) | | (0.22) | | (0.19) |
| Hansen | | 27.81 | | 28.26 |
| Test(p) | | (1.000) | | (1.000) |
| N | 450 | 420 | 450 | 420 |

Figures in parentheses are progressive t statistics. *, **, *** denote statistical significance levels at 10%, 5% and 1%, respectively.

### 5.2. Analysis of Regression Results in the Eastern, Central and Western Regions

With respect to the third part, the 30 provinces in China are classified into Eastern, Central and Western regions.

In order to better study whether there is a regional difference in the impact of urbanization on carbon emission efficiency, we use the dynamic spatial panel model to conduct regressions on the panel data of the three respective regions. The regression results are shown in Table 6.

**Table 6.** Regression results of the eastern, central and western regions.

| Variables | Eastern Region | Western Region | Central Region |
|---|---|---|---|
| | (Spatial-GMM) | (Spatial-GMM) | (Spatial-GMM) |
| $\tau$ (time lag term) | 0.145 *** | 0.136 *** | 0.127 *** |
| | (3.417) | (3.286) | (3.069) |
| $\rho$ (Spatial lag term) | 0.017 *** | 0.015 ** | 0.011 * |
| | (2.673) | (2.147) | (1.775) |
| Urban | 0.742 | 0.796 ** | 0.883 *** |
| | (1.406) | (2.185) | (2.927) |
| Population | 0.286 *** | 0.242 *** | 0.211 *** |
| | (4.741) | (4.170) | (3.635) |
| Pgdp | −0.039 | −0.050 ** | −0.058 *** |
| | (−1.507) | (−2.173) | (−2.619) |
| Structure | −0.762 *** | −0.711 ** | −0.675 |
| | (−2.614) | (−2.175) | (−1.490) |
| Energy | −1.021 *** | −1.067 *** | −0.086 *** |
| | (−5.186) | (−5.364) | (−5.027) |
| _cons | −1.906 *** | −2.052 *** | −2.136 *** |
| | (−7.423) | (−7.658) | (−8.035) |
| AR(1) | −3.46 | −3.74 | −3.62 |
| Test(p) | (0.001) | (0.000) | (0.000) |
| AR(2) | −1.25 | −1.29 | −1.22 |
| Test(p) | (0.21) | (0.20) | (0.24) |
| Hansen | 23.25 | 24.41 | 25.07 |
| Test(p) | (1.000) | (1.000) | (1.000) |
| N | 165 | 165 | 120 |

Figures in parentheses are progressive t statistics. *, **, *** denote statistical significance levels at 10%, 5% and 1%, respectively.

Table 6 shows that the dynamic lag term of carbon emission efficiency is significantly positive, indicating that there are significant dynamic effects of carbon emission efficiency in the three regions. Comparing the coefficients, we see that the Eastern region is the largest, the Central region is the second largest and the Western region is the smallest. This may be due to the higher carbon emission efficiency in the Eastern region which depends more on the early stage. The spatial lag term of carbon emission efficiency is also significantly positive which shows that there are significant spatial effects in three regions. Comparing the coefficients, the Eastern region is the largest, the Central region is the second and the Western region is the smallest. This may be due to the higher level of economic development in the Eastern region; the closer economic exchanges and ties between regions makes the spatial effect of carbon emission efficiency stronger.

From the regression results in Table 6, there are indeed regional differences in the impact of urbanization on carbon emission efficiency. Specifically, urbanization has no significant effect on carbon emission efficiency in the Eastern region. Urbanization is significantly conducive to improvements in carbon emission efficiency in the Central and Western regions with its effects in the Western region stronger. This is mainly because urbanization has a marginal decreasing effect on the improvement of carbon emission efficiency and the urbanization level in Eastern China has entered the late stage. It is difficult for the urbanization rate to further increase significantly, so its promoting effect is not significant. The Central and Western regions are still in the stage of continuous development of urbanization. The intensive use of resources and the accumulation of human capital brought by it can significantly promote the efficiency of carbon emissions. Compared with the Central region, the level of urbanization in the Western region is lower, which makes its effect stronger.

## 6. Conclusions

Based on panel data of 30 provinces in China from 2000 to 2015, this paper first calculates carbon emissions and then uses the DEA model to calculate the carbon emission efficiency. Thereafter, it employs both static and dynamic panel models to comprehensively investigate those factors affecting carbon emission efficiency. The main conclusions are as follows: (1) The carbon emission efficiency of China's provinces is on the rise but with a significant difference in the carbon emission efficiency between the Eastern and Western regions. The Eastern region has the highest carbon emission efficiency while the Western region has the lowest. (2) Urbanization can significantly improve carbon efficiency but there are significant regional differences in this contribution. (3) Population density has a significant positive impact on carbon emission efficiency but industrial structure, energy intensity and per capita GDP all have a significant negative impact on carbon emission efficiency.

Based on empirical test results, the government should vigorously promote the construction of new urbanization model with a focus on high-quality development. Such high-quality urbanization can improve carbon emission efficiency and ultimately make contributions to the overall reduction of carbon dioxide emissions. Different regions should adhere to different urbanization strategies. Considering the heterogeneity impact of urbanization scale, the Eastern region with its higher level of urbanization should accelerate the coordination of urbanization development levels in the other regions. The Eastern provinces have more developed economies and higher levels of technology. They should change their traditional model of economically extensive urbanization and instead promote both high-quality and coordinated action as regards regional agglomeration urbanization. The Central and Western regions should continue to accelerate the promotion of urbanization in Central cities. They should encourage the transfer from large and medium-sized cities in some provinces (cities) to smaller cities, forming an urbanization development model that is both mutually advantageous and cooperates in the region to reduce carbon emissions.

**Author Contributions:** L.L. (Lianshui Li) proposed and implemented the study; Y.C. provided the data and analyzed the data; and L.L. (Lianshui Li), Y.C. and L.L. (Liang Liu) wrote the paper. All authors have read and agreed to the published version of the manuscript.

**Funding:** The work in this paper was supported by the National Natural Science Foundation of China (grant no. 71673145), the Report Project on the Development of Philosophy and Social Sciences of China's Ministry of Education (grant no. 13JBG004).

**Conflicts of Interest:** The authors declare no conflict of interest.

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
