# Peer review of "Research on the Effect of Urbanization on China’s Carbon Emission Efficiency"

_sustainability, doi:10.3390/su12010163_

Round 1

Reviewer 1 Report

The paper “Research on the Effect of Urbanization on China's Carbon Emission Efficiency” is interesting.

General Comments

The paper “Research on the Effect of Urbanization on China's Carbon Emission Efficiency” is interesting.

This paper selects panel data of 30 provinces in China from 2000 to 2015, evaluates the carbon emission efficiency of each province with DEA method, and empirically works on the effect of urbanization on carbon emission efficiency based on STIRPAT's expansion form. The results show the carbon emission efficiency of China's provinces showed an upward trend, and the carbon emission efficiency of the eastern, central and western regions was significantly different, with the highest in the eastern region, the second in the central region, and the lowest in the western region. After controlling population density, economic development level, energy intensity and industrial structure, urbanization can significantly improve carbon emission efficiency, but there are regional differences in this promotion effect. Urbanization is conducive to the improvement of carbon emission efficiency in the central and western regions, where the promotion effect of the western region is stronger, but the promotion effect of the eastern region is not significant. Based on the conclusions above, this paper puts forward policy recommendations to promote China's low carbon and environmental protection.

2.The paper should be cite some newest literatures in the introduction to shows the importance of carbon emission reduction and other environmental pollution reduction. For example, as the following literatures.

    Mojie Li, Zhifu Mi, D'Maris Coffman, Yi-Ming Wei, Assessing the policy impacts on non-ferrous metals industry's CO2 reduction: Evidence from China, Journal of Cleaner Production, Volume 192, 10 August 2018, Pages 252-261

    Zeng S, Jiang, C., Ma, C., Su, B., Investment Efficiency of the New Energy Industry in China, Energy Economics, 2018, 70:536-544.

    Mingxing Sun, Yutao Wang, Lei Shi, Jiří Jaromír Klemeš. Uncovering energy use, carbon emissions and environmental burdens of pulp and paper industry: A systematic review and meta-analysis. Renewable and Sustainable Energy Reviews, 2018 (92): 823-833.

The extensive editing of English language and style is required.

Author Response

Reviewer 1:

The paper “Research on the Effect of Urbanization on China's Carbon Emission Efficiency” is interesting.

1.This paper selects panel data of 30 provinces in China from 2000 to 2015, evaluates the carbon emission efficiency of each province with DEA method, and empirically works on the effect of urbanization on carbon emission efficiency based on STIRPAT's expansion form. The results show the carbon emission efficiency of China's provinces showed an upward trend, and the carbon emission efficiency of the eastern, central and western regions was significantly different, with the highest in the eastern region, the second in the central region, and the lowest in the western region. After controlling population density, economic development level, energy intensity and industrial structure, urbanization can significantly improve carbon emission efficiency, but there are regional differences in this promotion effect. Urbanization is conducive to the improvement of carbon emission efficiency in the central and western regions, where the promotion effect of the western region is stronger, but the promotion effect of the eastern region is not significant. Based on the conclusions above, this paper puts forward policy recommendations to promote China's low carbon and environmental protection.

2.The paper should be cite some newest literatures in the introduction to shows the importance of carbon emission reduction and other environmental pollution reduction. For example, as the following literatures.

  Mojie Li, Zhifu Mi, D'Maris Coffman, Yi-Ming Wei, Assessing the policy impacts on non-ferrous metals industry's CO2 reduction: Evidence from China, Journal of Cleaner Production, Volume 192, 10 August 2018, Pages 252-261

  Zeng S, Jiang, C., Ma, C., Su, B., Investment Efficiency of the New Energy Industry in China, Energy Economics, 2018, 70:536-544.

  Mingxing Sun, Yutao Wang, Lei Shi, Jiří Jaromír Klemeš. Uncovering energy use, carbon emissions and environmental burdens of pulp and paper industry: A systematic review and meta-analysis. Renewable and Sustainable Energy Reviews, 2018 (92): 823-833.

The extensive editing of English language and style is required.

Response:

Thanks a lot for the reviewer’s comments about the contributions of this paper. We have cited some newest literature in the introduction to show the importance of carbon emissions by highlight the major changes using red text (page 1-2). Meanwhile, we have also looked for a native speaker to edit the English language and style.

Reviewer 2 Report

This paper estimates the impact of the urbanization rate on CO2 emissions in China.  The topic and results are of clear and obvious importance.  The analysis is well done, using an appropriate methodology.

My main comments are as follows:

The paper connects to a few more literatures than the brief lit review indicates. I think bringing more related topics in, at the beginning of the paper, could broaden the audience.  For example, many recent papers have begun to think about population demographics as a source of variation in pollution (more on this below, but see Lugauer et al (2014) and Zagheni (2011), and the subsequent papers that cite these). The connection to abatement efforts in China could be discussed. See Wei et al. (2012), for example. The ‘Environmental Kuznets Curve’ literature also seems relevant. Fei et al. (2017) also should be cited. The idea of ‘pollution offshoring’ (even if it is all internal to China) also seems relevant. See Ederington et al. Related to point 1a, people of different ages and demographics consume different things and have different migration patterns. In particular, prime age individuals work and drive more.  The urbanization patterns documented in this paper could be related.  Indeed, the urbanization rate might be highly correlated with the age structure.  I think this point deserves more discussion.  I could even imagine a quick test by replacing the POP variable with a more detailed set of demographic controls.  If the results (coefficient estimate on urbanization) diminish, it does not necessarily gainsay the paper’s results, though.  It would just show that the urbanization effect works in conjunction (or through) who it is that is ‘urbanizing’ an area.

My other points are more minor, but also worth considering:

In the SBM model part, what is the definition of the “dmu”? What is lambda in model 1? I speculate that formula 2 should also include t, or at least this possibility could be discussed.  Figure 5 indicates that Hebei has a very low carbon efficiency. It might be worthwhile to show the yearly trend.   Does Hebei’s carbon emission efficiency significantly decline in the recent decade, or have an upward trend more recently? Relatedly, many energy plants that were initially installed in Beijing were relocated to Hebei. Among the most famous was Shougang Group Co.,Ltd  (Capital Steel)’s factory, which was relocated to Hebei prior to 2008 Olympic Games due to environmental concerns. Reallocation of the plants could be a significant factor in influencing emissions. I like the first paragraph of the conclusion section. The rest seems a bit speculative and not necessarily supported by the actual analysis in the paper.  I would delete points 1-5, but I guess this should be up to the authors’ discretion.

References

Wei, Ni, and Du. 2012. “Regional Allocation of Carbon Dioxide Abatement in China” China Economic Review.

Lugauer, Jensen, and Sadler. 2014. “An Estimate of the Age Distribution’s Effect on Carbon Dioxide Emissions” Economic Inquiry.

Fei, Wang, Su, Jin, Wang and Zhang. 2017. “Decomposition Analysis of Carbon Emission Factors from Energy Consumption in Guangdong Province from 1990 to 2014” Sustainability.

Zagheni (2011), Demography, “The Leverage of Demographic Dynamics on CO2 Emissions: Does Age Structure Matter?”.

"Footloose and Pollution-Free," (Ederington, A. Levinson and J. Minier), Review of Economics and Statistics 87 (2005); 92-99.

Author Response

Reviewer 2:

This paper estimates the impact of the urbanization rate on CO2 emissions in China. The topic and results are of clear and obvious importance. The analysis is well done, using an appropriate methodology. My main comments are as follows:

The paper connects to a few more literatures than the brief lit review indicates. I think bringing more related topics in, at the beginning of the paper, could broaden the audience. For example, many recent papers have begun to think about population demographics as a source of variation in pollution (more on this below, but see Lugauer et al (2014) and Zagheni (2011), and the subsequent papers that cite these). The connection to abatement efforts in China could be discussed. See Wei et al. (2012), for example. The ‘Environmental Kuznets Curve’ literature also seems relevant. Fei et al. (2017) also should be cited. The idea of ‘pollution offshoring’ (even if it is all internal to China) also seems relevant. See Ederington et al. Related to point 1a, people of different ages and demographics consume different things and have different migration patterns. In particular, prime age individuals work and drive more. The urbanization patterns documented in this paper could be related. Indeed, the urbanization rate might be highly correlated with the age structure. I think this point deserves more discussion. I could even imagine a quick test by replacing the POP variable with a more detailed set of demographic controls.  If the results (coefficient estimate on urbanization) diminish, it does not necessarily gainsay the paper’s results, though. It would just show that the urbanization effect works in conjunction (or through) who it is that is ‘urbanizing’ an area.

My other points are more minor, but also worth considering:

In the SBM model part, what is the definition of the “dmu”? What is lambda in model 1? I speculate that formula 2 should also include t, or at least this possibility could be discussed.  Figure 5 indicates that Hebei has a very low carbon efficiency. It might be worthwhile to show the yearly trend. Does Hebei’s carbon emission efficiency significantly decline in the recent decade, or have an upward trend more recently? Relatedly, many energy plants that were initially installed in Beijing were relocated to Hebei. Among the most famous was Shougang Group Co.,Ltd  (Capital Steel)’s factory, which was relocated to Hebei prior to 2008 Olympic Games due to environmental concerns. Reallocation of the plants could be a significant factor in influencing emissions. I like the first paragraph of the conclusion section. The rest seems a bit speculative and not necessarily supported by the actual analysis in the paper. I would delete points 1-5, but I guess this should be up to the authors’ discretion.

References

Wei, Ni, and Du. 2012. “Regional Allocation of Carbon Dioxide Abatement in China” China Economic Review.

Lugauer, Jensen, and Sadler. 2014. “An Estimate of the Age Distribution’s Effect on Carbon Dioxide Emissions” Economic Inquiry.

Fei, Wang, Su, Jin, Wang and Zhang. 2017. “Decomposition Analysis of Carbon Emission Factors from Energy Consumption in Guangdong Province from 1990 to 2014” Sustainability.

Zagheni (2011), Demography, “The Leverage of Demographic Dynamics on CO2 Emissions: Does Age Structure Matter?”.

"Footloose and Pollution-Free," (Ederington, A. Levinson and J. Minier), Review of Economics and Statistics 87 (2005); 92-99.

Response:

Thanks a lot for reviewer’s comments about the contributions of this paper. we have answered reviewer’s questions individually:

(1) We have cited some literature in the introduction about the influencing factors of carbon emissions, e.g., economic growth, population, renewables energy, and environmental regulation, which have been highlight using red text (page 1-2).

(2) We think that the effect of the age structure of urbanization on carbon emissions is an interesting and real topic. We tried to add the regional population structure to the model. However, there are two difficulties: first, this paper uses the panel data of 30 provinces in China from 2000 to 2015, but National Bureau of Statistics of China (http://data.stats.gov.cn/easyquery.htm?cn=E0103) lacks the data of population structure in 2000, 2001 and 2010. Second, we tried to add population structure using the data of from 2011 to 2015, but the correlation coefficient between urbanization (using the proportion of interprovincial urban population to the total population) and population structure (using the proportion of aging 15-65 as a percentage of the total population ) is 0.6321, which may be multicollinearity. Therefore, our further research will focus on the effect of age structure of urbanization on carbon emissions, and thanks again for your comments.

(3) In the SBM model part, “dmu” is the abbreviation of the decision-making unit, which is the assessed unit (provinces or cities) in this paper.

(4) In model 1, “lambda” is the weight column matrix of input and output of the DMU.

(5) We have added the t factor in formula 2, thanks for reviewer’s comment.

(6) Table 2 and Figure 1-3 show the yearly trend of carbon emission efficiency. Based on the opinions of reviewers, we have described the change of Hebei’s carbon emission efficiency and explained the possible reason (page 7). 

(7) Reviewers provided a good comment on the conclusion. We have deleted points 1-5 in the conclusion and rewritten it.

(8) We have looked for a native speaker to edit English language and style.

Reviewer 3 Report

There are many papers published on energy efficiency in China regions based on DEA. What is the novelty of this paper? In addition, there is no discussion section in this paper. This section is necessary to compare and discuss results of this study with similar studies in this area. This is the main  and important limitation of this paper. I strongly recommend authors revise paper by stressing input. What is advantages of their methodology in comparison with similar studies in China. Policy implications need to be developed based on current policies to promote energy efficiency in China analysis. The conclusions are weak, they are too long and not well -structured. They should also include findings from discussion sections which is missing.

Author Response

Reviewer 3:

There are many papers published on energy efficiency in China regions based on DEA. What is the novelty of this paper? In addition, there is no discussion section in this paper. This section is necessary to compare and discuss results of this study with similar studies in this area. This is the main and important limitation of this paper. I strongly recommend authors revise paper by stressing input. What are advantages of their methodology in comparison with similar studies in China. Policy implications need to be developed based on current policies to promote energy efficiency in China analysis. The conclusions are weak, they are too long and not well-structured. They should also include findings from discussion sections which is missing.

Response:

Thanks a lot for the reviewer’s comments. We have revised the manuscript in the following three aspects:

(1) We have cited some newest literatures to show the importance of carbon emission reduction and the novelty of this paper, which have been highlight using red text (page 1-2).

(2) We have added the discussion section in page 3:

We think that the literature focuses on the effect of urbanization on carbon emissions and draws rich conclusions. This does indeed provide some reference for the study of this article. However, there is still little literature on the effect of urbanization on carbon emission efficiency. In the context of China's economic growth and the need for reduction in carbon emissions, improving carbon emission efficiency is of great importance. At the same time, an analysis of the impact of urbanization on carbon emission efficiency is certainly worthwhile as it can serve to guide the development of China's new urbanization. Compared to existing research, the innovations of this paper are as follows: first, this paper studies the relationship between urbanization and carbon emission efficiency, an important complement to the literature. Second, using both the static and dynamic spatial models, we estimate the space overflow effect of urbanization on carbon emission efficiency. Third, taking the case of China’s 30 provinces (cities), we compare the heterogeneous impacts of urbanization scale; this is important for both the formulation of urbanization development policies and science-based urban development in China.

(3) We have rewritten and shortened the conclusions:

Based on empirical test results, the government should vigorously promote the construction of new urbanization model with a focus on high-quality development. Such high-quality urbanization can improve carbon emission efficiency and ultimately make contributions to the overall reduction of carbon dioxide emissions. Different regions should adhere to different urbanization strategies. Considering the heterogeneity impact of urbanization scale, the Eastern region with its higher level of urbanization should accelerate the coordination of urbanization development levels in the other regions. The Eastern provinces have more developed economies and higher levels of technology. They should change their traditional model of economically extensive urbanization and instead promote both high-quality and coordinated action as regards regional agglomeration urbanization. The Central and Western regions should continue to accelerate the promotion of urbanization in Central cities. They should encourage the transfer from large and medium-sized cities in some provinces (cities) to smaller cities, forming an urbanization development model that is both mutually advantageous and cooperates in the region to reduce carbon emissions.

Round 2

Reviewer 1 Report

The article has been greatly improved 

Reviewer 3 Report

The authors have corrected manuscript and provided relevant answers to reviewers comments and paper can be published in current form.